# Genetic and clinical landscape of breast cancers with germline *BRCA1/2* variants

Yukiko Inagaki-Kawata[1,2], Kenichi Yoshida [1], Nobuko Kawaguchi-Sakita[3], Masahiro Kawashima[2], Tomomi Nishimura[1,2], Noriko Senda[2], Yusuke Shiozawa[1], Yasuhide Takeuchi[1,4,5], Yoshikage Inoue[1], Aiko Sato-Otsubo[1], Yoichi Fujii[1], Yasuhito Nannya[1], Eiji Suzuki[2], Masahiro Takada[2], Hiroko Tanaka [6], Yuichi Shiraishi[7], Kenichi Chiba[7], Yuki Kataoka[8], Masae Torii[9], Hiroshi Yoshibayashi[9], Kazuhiko Yamagami[10], Ryuji Okamura[11], Yoshio Moriguchi [12], Hironori Kato[13], Shigeru Tsuyuki[14], Akira Yamauchi[15], Hirofumi Suwa[16], Takashi Inamoto[17], Satoru Miyano[6,7], Seishi Ogawa [1,4,18✉] & Masakazu Toi [2✉]

The genetic and clinical characteristics of breast tumors with germline variants, including their association with biallelic inactivation through loss-of-heterozygosity (LOH) and second somatic mutations, remain elusive. We analyzed germline variants of 11 breast cancer susceptibility genes for 1,995 Japanese breast cancer patients, and identified 101 (5.1%) pathogenic variants, including 62 *BRCA2* and 15 *BRCA1* mutations. Genetic analysis of 64 *BRCA1/2*-mutated tumors including TCGA dataset tumors, revealed an association of biallelic inactivation with more extensive deletions, copy neutral LOH, gain with LOH and younger onset. Strikingly, *TP53* and *RB1* mutations were frequently observed in *BRCA1*- (94%) and *BRCA2*- (9.7%) mutated tumors with biallelic inactivation. Inactivation of *TP53* and *RB1* together with *BRCA1* and *BRCA2*, respectively, involved LOH of chromosomes 17 and 13. Notably, *BRCA1/2* tumors without biallelic inactivation were indistinguishable from those without germline variants. Our study highlights the heterogeneity and unique clonal selection pattern in breast cancers with germline variants.

[1] Department of Pathology and Tumor Biology, Kyoto University, Kyoto, Japan. [2] Department of Breast Surgery, Kyoto University, Kyoto, Japan. [3] Department of Clinical Oncology, Kyoto University Hospital, Kyoto, Japan. [4] Institute for the Advanced Study of Human Biology (WPI-ASHBi), Kyoto University, Kyoto, Japan. [5] Department of Diagnostic Pathology, Kyoto University, Kyoto, Japan. [6] Laboratory of Sequence Analysis, Human Genome Centre, Institute of Medical Science, The University of Tokyo, Tokyo, Japan. [7] Laboratory of DNA Information Analysis, Human Genome Center, Institute of Medical Science, The University of Tokyo, Tokyo, Japan. [8] Hospital Care Research Unit/Department of Respiratory Medicine, Hyogo Prefectural Amagasaki General Medical Center, Amagasaki, Japan. [9] Department of Breast Surgery, Japanese Red Cross Wakayama Medical Center, Wakayama, Japan. [10] Department of Breast Surgery, Shinko Hospital, Kobe, Japan. [11] Department of Breast Surgery, Yamatotakada Municipal Hospital, Yamatotakada, Japan. [12] Department of Breast Surgery, Kyoto City Hospital, Kyoto, Japan. [13] Department of Breast Surgery, Kobe City Medical Center General Hospital, Kobe, Japan. [14] Department of Breast Surgery, Osaka Red Cross Hospital, Osaka, Japan. [15] Department of Breast Surgery, Kitano Hospital, Osaka, Japan. [16] Department of Breast Surgery, Hyogo Prefectural Amagasaki General Medical Center, Amagasaki, Japan. [17] Tenri Health Care University, Tenri, Japan. [18] Department of Medicine, Centre for Haematology and Regenerative Medicine, Karolinska Institute, Stockholm, Sweden. ✉email: sogawa-tky@umin.ac.jp; toi@kuhp.kyoto-u.ac.jp

Germline predisposition plays a substantial role in breast cancer, the most prevalent cancer in women. Management and prevention of breast cancer would, therefore, benefit from better knowledge and understanding of the genetic cause behind such familial predisposition[1]. Previous studies reported that pathogenic germline mutations account for 10.7% of breast cancer cases in a Western cohort[2] and 9.2% of those in a Chinese cohort[3]. Breast cancer is also prevalent in Japan, affecting 116.3 per 100,000 women, where germline predisposition has been confirmed or suspected in as many as 5.7% of cases[4]. However, genetic studies of germline mutations that result in a predisposition to breast cancer are limited in the Japanese population[4–6]. In particular, the effects of pathogenic germline variants on somatic mutations and clinical/pathological phenotypes of accompanying breast cancers are poorly understood. Exception is the well-established mutational signatures associated with germline mutations in *BRCA1/2* and *PALB2*[7,8], which are key genes in DNA repair by homologous recombination (HR) of DNA double strand breaks[9]. Although a previous study analyzed tumors with germline *BRCA1/2* mutations in terms of presence or absence of biallelic inactivation[10], its genetic and clinical impact on breast cancers have not been fully elucidated.

In this study, therefore, we investigated pathogenic germline variants in 11 genes implicated in hereditary breast cancer, which were *BRCA1*, *BRCA2*, *TP53*, *PTEN*, *CDH1*, *STK11*, *NF1*, *PALB2*, *ATM*, *CHEK2*, and *NBN*[1,11–18], for 1995 unselected Japanese women with breast cancer, using targeted-capture sequencing of pooled DNA (Supplementary Fig. 1a). For those patients for whom tumor samples were available, the somatic alterations in the tumor were also interrogated in order to link the genetic features of the germline risk alleles to the associated tumor clinical presentations. In particular, we investigated the effects of biallelic inactivation of *BRCA1/2* genes on the somatic mutations, and copy number (CN) abnormalities (CNAs) and clinical features of the resulting breast cancers.

## Results

**Prevalence of germline mutations**. The characteristics of the study participants are shown in Table 1. Of the 1995 patients analyzed, 101 (5.1%) carried pathogenic germline variants, comprising 54 (53.5%) frameshift insertion or deletions (indels), 35 (34.7%) nonsense mutations, 7 (6.9%) missense mutations, 4 (4.0%) splice site mutations, and 1 (1.0%) synonymous mutation (Supplementary Fig. 1b). With respect to the 11 genes of interest, 77 of the total (3.9%) patients carried a variant in *BRCA2* (n = 62, 3.1%) or *BRCA1* gene (n = 15, 0.8%), followed by *PALB2* (n = 9, 0.5%), *TP53* (n = 4, 0.2%), *PTEN* (n = 4, 0.2%), *CHEK2* (n = 3, 0.2%), *ATM* (n = 3, 0.2%), and *NF1* (n = 1, 0.05%) (Fig. 1a, Supplementary Table 1). No pathogenic variants were identified in *CDH1*, *STK11*, and *NBN*. None of the patients carried two or more pathogenic variants. The genetic mutations were widely distributed along the entire coding region of the genes, except for missense mutations on the DNA binding domain of *TP53* (Supplementary Fig. 2). Some variants were detected in ≥2 patients, such as L63X in *BRCA1*[19] and c.5576_5579del in *BRCA2*[3,4], which have been previously suggested as founder mutations in the Asian population[19,20].

**Characteristics of patients carrying germline variants**. The profiles of patients carrying germline variants in each gene are shown in Supplementary Table 2. Pathogenic variants were more frequently identified in patients with a family history of breast cancer (n = 41, 11.0%), compared with those without (n = 50, 3.4%) (P < 0.00001). Of the analyzed genes, *BRCA2* was the most frequently mutated in both patients with and without a family

**Table 1 Characteristics of 1995 patients enrolled in this study.**

|  | Number of patients |
|---|---|
| *Age* |  |
| ~35 | 51 |
| 36–45 | 317 |
| 46–55 | 461 |
| 56–65 | 522 |
| 66 | 644 |
| *Histology* |  |
| IDC | 1484 |
| ILC | 74 |
| DCIS | 212 |
| Others | 106 |
| *Phenotype* |  |
| ER+ | 1540 |
| ER− | 377 |
| HER2+ | 310 |
| HER2− | 1382 |
| Ki67 high (>14) | 967 |
| Ki67 low | 721 |
| *Histological grade* |  |
| 3 | 146 |
| 2 | 419 |
| 1 | 217 |
| *Clinical stage* |  |
| 0 | 254 |
| I | 774 |
| II | 733 |
| III | 142 |
| IV | 53 |
| Unknown | 39 |
| *Past history* |  |
| Ovarian cancer | 14 |
| *Family history*[a] |  |
| Breast cancer | 348 |
| Ovarian cancer | 33 |

Abbreviations: *IDC* Invasive ductal carcinoma, *ILC* invasive lobular carcinoma, *DCIS* ductal carcinoma in situ.
[a]Family history refers to reported breast or ovarian cancer in first or second-degree relative.

history (Supplementary Table 3). A quarter of the patients with germline variants did not fulfill the NCCN criteria[21] for assessment as high-risk for genetic or familial cancers.

The median age at diagnosis of patients with pathogenic germline variants was 53 years, which was younger than that of patients with no pathogenic variants (60 years) (P < 0.00001) (Fig. 1b, Supplementary Fig. 3a). *BRCA1* germline mutations were associated with younger age at diagnosis (median age, 43 years), compared with *BRCA2* (median age, 56 years; P = 0.08), and other 6 genes (median age, 52 years; P = 0.08). Among early onset breast cancer patients who were diagnosed before 35 years of age, prevalence of germline variants in *BRCA1*, *BRCA2*, and other genes was 9.8%, 17.1%, and 4.9%, respectively (Supplementary Fig. 3b).

As previously reported[3], *BRCA1* variant carriers were more likely to have triple-negative (estrogen receptor (ER) negative, progesterone receptor (PR) negative, and HER2 negative) diseases, compared with those with other germline variants (P = 0.0007) and those without germline mutations (P = 0.0001) (Fig. 1c). In contrast, there was no obvious tumor subtype associated with *BRCA2* variant carriers. Advanced (T2–T4) breast cancers were more common in *BRCA1*-mutated cases, compared with those with pathogenic germline mutations in other genes (P = 0.08) or those with no pathogenic variants detected (P = 0.05) (Fig. 1d), especially among younger patients

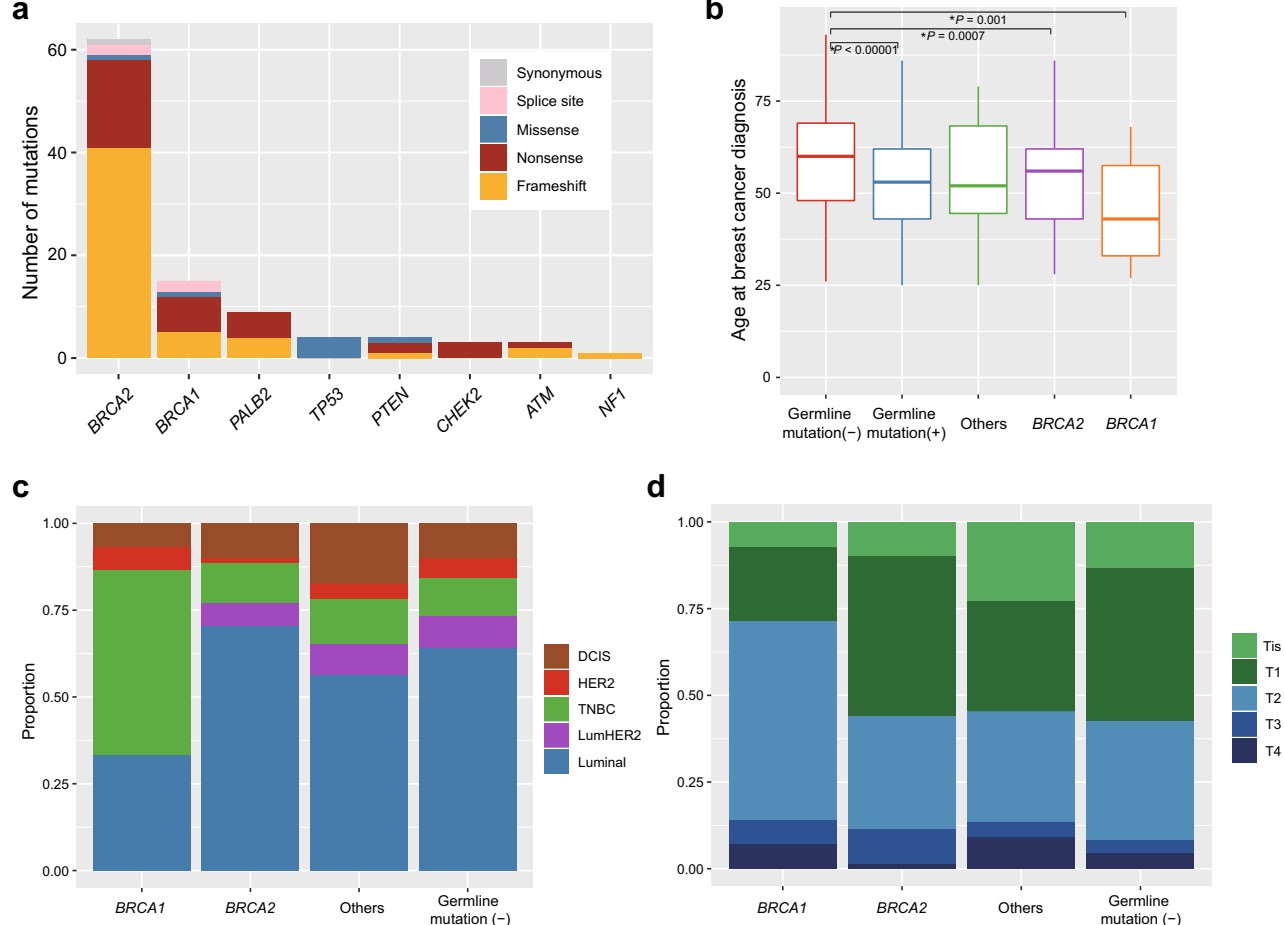

**Fig. 1 Clinical characteristics of breast tumors harboring pathogenic germline variants. a** Numbers and types of germline variants identified in each gene. **b** Box-and-whisker plots of age at breast cancer diagnosis for tumors with germline variants (n = 101), including BRCA1 (n = 15), BRCA2 (n = 62), and others (n = 24), and those without (n = 1892). The boxes indicating median and interquartile range, and the whiskers denoting the range. Asterisks indicate significance difference (Mann–Whitney U test: P < 0.05). **c** Distribution of phenotype according to the status of germline mutations. DCIS ductal carcinoma in situ, TNBC triple-negative breast cancer, LumHER2 luminalHer2. **d** Distribution of t factor for tumors according to the status of germline mutations. Tis carcinoma in situ.

(Supplementary Fig. 3c). Lymph node involvement did not significantly correlate with the status of pathogenic germline variants (Supplementary Fig. 3d). Survival data were available for 1387 (69.5%) of 1995 patients with a median follow-up of 3 years (range: 0–35 years). No prognostic impact of germline mutations was demonstrated for overall and disease-free survival in both univariate and multivariate analyses (Supplementary Fig. 4, Supplementary Table 4).

**Somatic alterations in tumors with germline BRCA1 and BRCA2 variants**. Tumor samples were obtained from 30 patients with pathogenic germline variants in BRCA2 (n = 25) and BRCA1 (n = 5), as well as from an additional 30 patients without pathogenic germline mutations. Somatic mutations in common breast cancer drivers and CNAs were analyzed for these samples using targeted panel sequencing (Supplementary Table 5). In total, 19 of 30 samples with germline variants in BRCA1/2 had one or more somatic mutations in 18 driver genes with a median of 1 mutation/sample, which was significantly smaller than those without germline variants (median 2 mutations/sample, P = 0.004) (Fig. 2a, Supplementary Table 6). For tumors with germline BRCA1/2 mutations, somatic mutations were most frequently detected in PIK3CA (n = 6), TP53 (n = 6) and KMT2C (n = 6) (Fig. 2b, Supplementary Table 7). All samples had CNAs,

regardless of the presence or absence of a pathogenic germline mutation. Even though the two-hit hypothesis of tumorigenesis predicts that majority of cases will have biallelic inactivation of the relevant cancer predisposing loci, biallelic inactivation of the predisposing alleles was found in only 20 cases (67%), while the remaining 10 retained an intact allele (mono-allelic inactivation). For each of the 20 cases, biallelic inactivation was caused by loss-of-heterozygosity (LOH) affecting the relevant germline variant loci. Indeed, nearly all cases involved deletions in BRCA2 (17/17) and BRCA1 (1/3), followed by copy-neutral LOH (uniparental disomy) (n = 1) and gain with LOH (n = 1). These results suggest that LOH is the predominant mechanism of biallelic inactivation of BRCA1/2 genes. Interestingly, one tumor with a germline BRCA2 variant also harbored a low allele frequency of somatic BRCA2 mutation in addition to LOH, suggesting clonal heterogeneity in the tumor over time, with independent events leading to biallelic inactivation.

**Mono-allelic vs. biallelic inactivation of BRCA1/2**. BRCA1/2 genes normally function in DNA repair and their deleterious mutation has been linked to HR deficiency. Hence, in our evaluation of tumors with mono-allelic and biallelic inactivation of BRCA1/2, we first analyzed CNAs (Fig. 2c). As seen for representative cases in Fig. 2d, samples with mono-allelic BRCA1/2

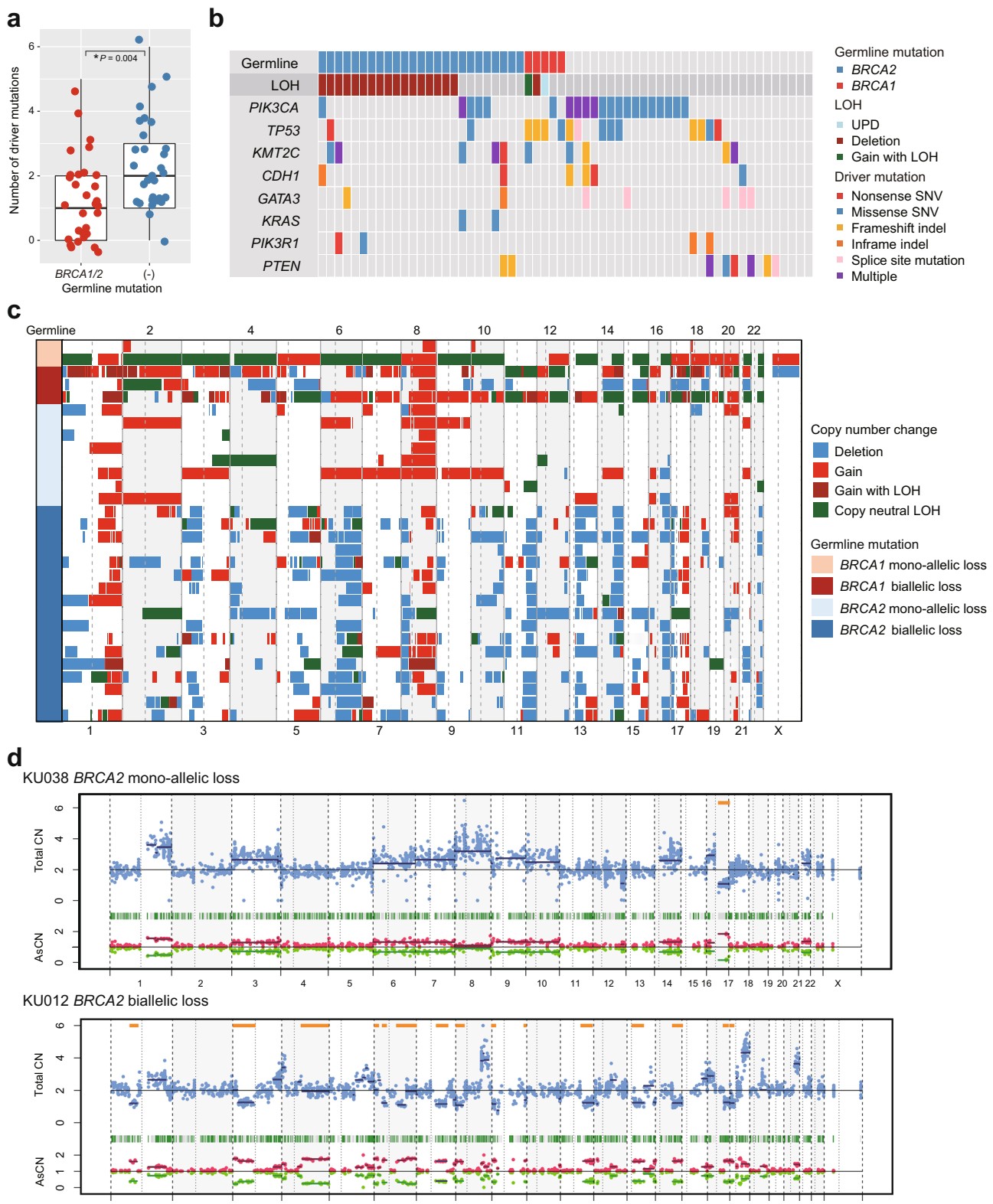

**Fig. 2 Genetic landscape of breast cancers with pathogenic germline variants in *BRCA1/2*. a** Numbers of driver mutations in tumors with germline variants in *BRCA1/2* (*n* = 30) and those without (*n* = 30). Asterisks indicate significance difference (Mann–Whitney *U* test: *P* < 0.05). **b** The landscape of somatic variants in tumors with germline variants in *BRCA1/2* (*n* = 30) and those without (*n* = 30). Distribution and types of recurrently mutated genes in tumors with germline *BRCA1/2* mutations are shown. **c** Genome-wide CN changes in tumors with germline variants in *BRCA1/2*. Red: gain, dark red: gain with LOH, green: copy neutral LOH, blue: deletion. **d** Representative CN plots of tumors with and without biallelic inactivation. Regions with LOH are shown in orange lines. CN copy number, AsCN allele specific CN.

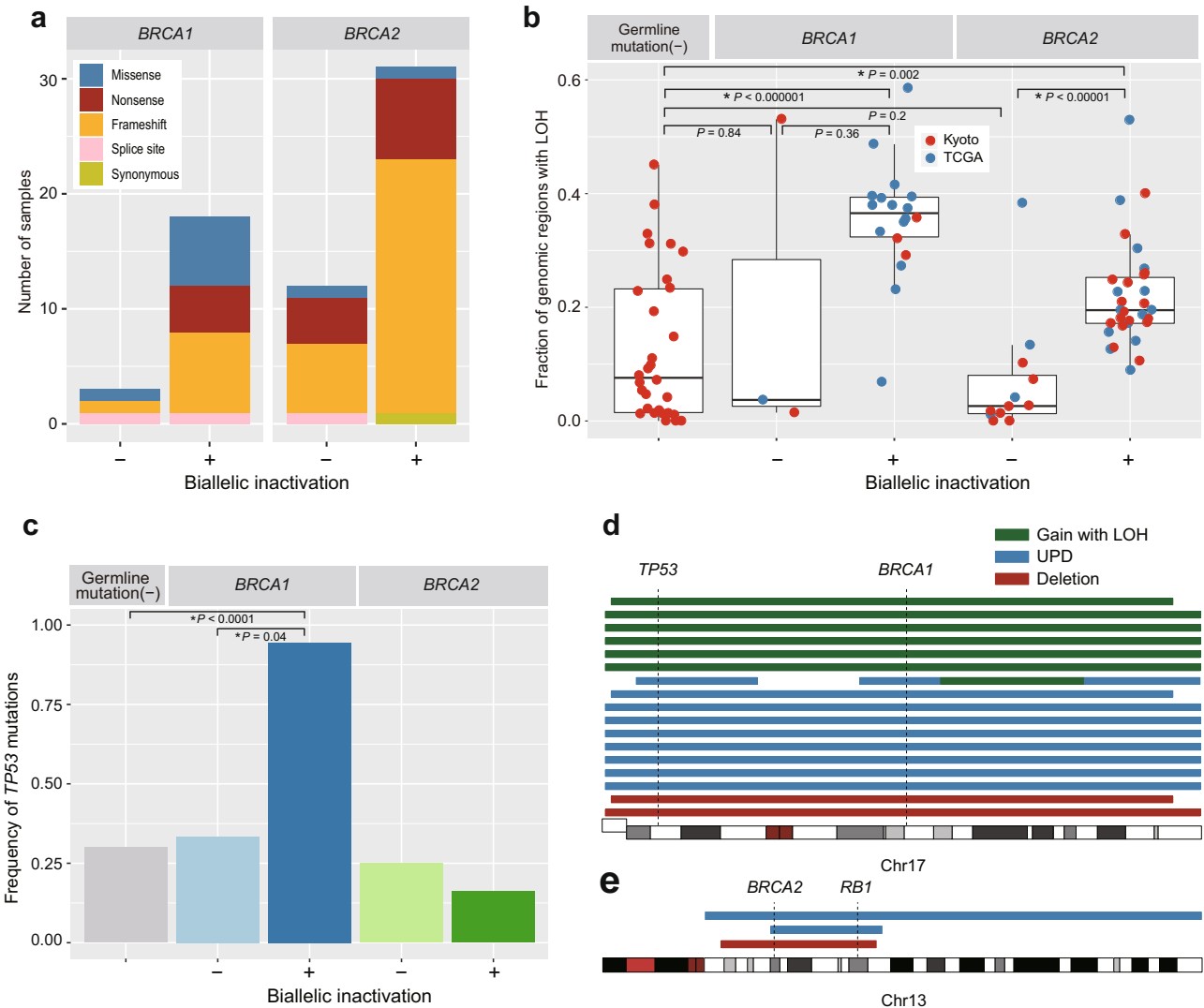

**Fig. 3 Genetic characteristics of breast tumors harboring germline *BRCA1/2* mutations with and without biallelic inactivation. a** Types of germline variants in *BRCA1/2* according to the status of biallelic inactivation ($n = 64$). **b** Fraction of genomic regions with LOH in tumors without germline variants ($n = 30$) and in those harboring germline *BRCA1* mutations with ($n = 18$) and without ($n = 3$) biallelic inactivation and those with germline *BRCA2* mutations with ($n = 31$) and without ($n = 12$) biallelic inactivation. Asterisks indicate significance difference (Mann–Whitney *U* test: $P < 0.05$). **c** Frequency of *TP53* mutations in tumors without germline variants ($n = 30$) and those harboring germline *BRCA1/2* mutations with and without biallelic inactivation ($n = 64$). Asterisks indicate a significance difference (Fisher test: $P < 0.05$). **d** LOHs on chromosome 17 in patients with germline *BRCA1* mutations ($n = 17$). LOHs identified in a single patient are shown on a line. **e** LOHs on chromosome 13 in patients with germline *BRCA2* mutations ($n = 3$).

lesions mainly showed CN gains with rare deletions. In contrast, biallelic *BRCA1* inactivation showed extensive gains with or without LOH, and those with biallelic *BRCA2* inactivation were characterized by extensive deletions or copy neutral LOHs. Although no specific patterns of CNAs have been reported in breast cancers with biallelic *BRCA1/2* inactivation, these observations led us to hypothesize that specific genomic rearrangements caused by HR deficiency could cause characteristic CN changes. Combining the Japanese cases with 34 cases from TCGA breast cancer cases, we tested this hypothesis using a total of 64 cases with germline *BRCA1/BRCA2* variants. In total, biallelic inactivation of *BRCA1/2* loci was observed in 49 of 64 (77%) patients: 18/21 (86%) cases with *BRCA1* and 31/43 (72%) cases with *BRCA2* germline mutations (Fig. 3a). Tumors with mono-allelic *BRCA1* and *BRCA2* mutations are less common in the TCGA cases, compared with the Japanese cases, particularly those with mono-allelic *BRCA1* mutations (1/16 vs. 2/5 for *BRCA1* and 4/18 vs. 8/25 for *BRCA2*). The predominant role of LOH was also

confirmed in TCGA cases, in which LOH explained 93% of biallelic inactivation in both *BRCA1*- and *BRCA2*-mutated cases, whereas biallelic inactivation via compound germline and somatic mutations were found only in two cases.

We next investigated characteristic patterns of mutations and structural variants (SVs) associated with biallelic *BRCA1* or *BRCA2* inactivation, focusing on mutational signatures and SVs using whole-exome sequencing (WES) data in the TCGA cohort. Four predominant mutational signatures were identified using pmsignature[22] (Supplementary Fig. 5a). Of these, the mutational signature caused by deficient HR (Sig_3) was more frequent in tumors with biallelic *BRCA2* inactivation than those without germline mutations, which concurs with previous reports[7,8] (Supplementary Fig. 5b, c). In analysis of SVs for tumors with biallelic inactivation of *BRCA1* and *BRCA2*, compared to tumors without germline variants, increased occurrence of tandem duplications and deletions (for *BRCA1* inactivation) and deletions (for *BRCA2* inactivation) were observed (Supplementary Fig. 5d,

e). By contrast, tumors with mono-allelic inactivation of either *BRCA* gene did not show an increase in Sig_3 mutations and deletions/tandem duplications. In addition, tumors with biallelic *BRCA2* inactivation exhibited significantly more extensive LOH lesions compared with those with mono-allelic inactivation and those without germline variants (Fig. 3b). Tumors with biallelic *BRCA1* inactivation also tended to have more extensive LOH than those without germline variants, but a comparison with mono-allelic and biallelic *BRCA1* inactivation was inconclusive due to the small number of patients with tumors of this mono-allelic category. These results suggest that biallelic *BRCA1/2* inactivation causes extensive CNAs, in addition to small SVs.

Strikingly, except for one case, which displayed compound germline and somatic *BRCA1* mutations, all but one tumors with biallelic *BRCA1* inactivation (17/18) harbored *TP53* mutations (Fig. 3c, Supplementary Fig. 5f). The *TP53* mutations were accompanied by high variant allele frequency and loss of an intact chromosome 17, leading to biallelic *TP53* inactivation (Fig. 3d, Supplementary Fig. 6a). Of added interest in this regard is the observation that some tumors with biallelic *BRCA2* inactivation, commonly accompanied by LOH of chromosome 13, also exhibited concomitant biallelic inactivation of *RB1*, which was mutated in 3 cases (Fig. 3e, Supplementary Fig. 6b). *RB1* mutations were more frequent in tumors with biallelic loss of *BRCA2* (3/31, 9.7%) than those without (22/858 cases in our cohort and TCGA dataset, 2.6%) ($P = 0.05$). These results suggest that loss of chromosomes 17 and 13 play an important role in the development of breast cancer in patients with mutated *BRCA1* and *BRCA2*, though inactivating *TP53* and *RB1*, respectively. However, the number of tumors identified in our study that exhibited *RB1* mutation in addition to biallelic loss of *BRCA2* is small ($n = 3$), and further studies are warranted to confirm the association of these genetic lesions.

Finally, we evaluated the clinical characteristics of patients with mono-allelic and biallelic *BRCA1/2* inactivation. Patients with biallelic inactivation showed a significantly younger onset than those without (median age at diagnosis: 47 vs. 59.6 years) ($P = 0.01$) (Fig. 4a), with no significant difference between patients with biallelic *BRCA1* and *BRCA2* inactivation. Although not significant, tumors having biallelic *BRCA1* inactivation tended to have more advanced (T2–T4) ($P = 0.09$) and triple-negative breast cancer ($P = 0.55$) (Fig. 4b, c, Supplementary Table 8). We also analyzed tumors with germline *BRCA1* and *BRCA2* variants for classification into PAM50 gene expression subtypes using TCGA samples. In accordance with previous reports, tumors with biallelic *BRCA1* inactivation were more frequently classified as basal-type[23,24], compared to those without (Fig. 4d). In strong contrast to biallelic BRCA1/2 inactivation, samples with mono-allelic *BRCA1/2* inactivation were not associated with younger age at onset, or an increase in triple-negative or basal-type tumors (Supplementary Table 8). The mutation status of *BRCA1/2* or the presence or absence of biallelic involvement of these genes did not affected overall or disease-free survival both in univariate and multivariate regression analyses (Supplementary Fig. 7, Supplementary Table 9).

## Discussion
In the current study, pathogenic germline mutations were detected in 5.1% of 1995 unselected Japanese breast cancer patients, which was equivalent to the frequency in the previous report of Japanese cohort[4]. The incidence rate of *BRCA1/2* variants was also relatively similar with those reported in Japanese patients[4] and Chinese population[3]. Given that the current study was restricted to detection of SVs and other variants that are not registered in ClinVar, the actual prevalence of pathogenic

germline variants might be underestimated. Importantly, a half of the cases (50/101) was negative for a family history of breast cancer, or did not fulfill the NCCN criteria for high-risk of familial cancer, indicating the importance of investigating germline DNA, even among sporadic breast cancer patients.

In line with the previous reports[8,10,25], tumors with mono-allelic *BRCA1* and *BRCA2* mutations were frequently observed in our cohort. Tumors with biallelic *BRCA1/2* inactivation had unique genetic features, in terms of CNA and *BRCA*-associated mutational signature and SVs, which were not seen in those with mono-allelic inactivation. Although the only significant clinical difference between tumors with mono-allelic and biallelic *BRCA1/2* inactivation was age at onset, the frequency of advanced stage, triple-negative or basal tumors tended to be higher in tumors with biallelic inactivation. The correspondence of biallelic *BRCA1/2* inactivation with earlier age of onset conflicts with results of a previous study[10]. Although both studies included TCGA samples, here, we carefully removed low quality samples and whole genome-amplified samples from the analysis. Furthermore, we considered compound germline and somatic mutations of *BRCA1/2* as contributors to biallelic inactivation, which likely increased the sensitivity of the dataset to detection of a correlation between the status of *BRCA1* and *BRCA2* and the clinical variables (Supplementary Fig. 8).

Tumors with mono-allelic *BRCA2* mutations and those without *BRCA1/BRCA2* mutations did not differ in their clinical presentation or analyses of additional genetic effects. In particular, mono-allelic tumors did not show an enhanced *BRCA*-related mutational signature or increase in SVs, which was seen in tumors with biallelic *BRCA1/2* inactivation. Nevertheless, mono-allelic germline *BRCA1/2* mutations show a significant enrichment in breast cancer patients, compared with the general population. Despite having only sequenced a portion of the tumors with *BRCA1/2* variants in our cohort (30/77), a mono-allelic loss-of-function mutation in *BRCA1* and *BRCA2* was more frequent than within the control cohort[4]: ≥2/1995 (0.1%) vs. 5/11,241 (0.04%) for *BRCA1* and ≥8/1995 (0.4%) vs. 15/11,241 (0.13%) for *BRCA2*. Thus, mono-allelic mutation alone does seem to play a role in the development of breast cancer, which is supported by several biological studies showing the effects of haploinsufficiency of *BRCA1/2* in carcinogenesis[26,27]. This raises the question as to whether or not platinum[28] or poly (ADP-ribose) polymerase (PARP) inhibitor[29] are also effective against tumors with mono-allelic mutations. A lower sensitivity of PARP inhibitors to cells with heterozygous *BRCA* mutations have indeed been reported using in vitro[30] and mouse models[31]. In contrast to this, however, Jonsson et al.[32] reported that the allelic status did not affect the response of in tumors with germline *BRCA1/2* mutations to PARP inhibitors[32]. Further evaluation of these drugs in the context of biallelic inactivation, is required in the future, incorporating clinical follow-up of patients, or clinical trials.

Finally, our study has revealed an intriguing linked mechanism of biallelic inactivation of *TP53* with *BRCA1* and *RB1* with *BRCA2*, respectively. Tumors showing biallelic inactivation of both *BRCA1* and *TP53* genes were almost invariably associated with loss of normal chromosome 17. Of interest, a previous study using fluorescence in situ hybridization (FISH) and immunohistochemistry of *TP53* on tumors with biallelic inactivation of both *BRCA1* and *TP53* have demonstrated that *TP53* mutation occurs before LOH of the intact chromosome 17 as there remained cells with two chromosome 17 alleles and mutated *TP53*. The subsequent biallelic *BRCA1* inactivation by LOH thus leads to biallelic inactivation of both *TP53* and *BRCA1* simultaneously[25]. A similar scenario is suggested here for *RB1* and *BRCA2* on chromosome 13, leading to simultaneous biallelic inactivation of the two genes via a deletion of part of chromosome 13.

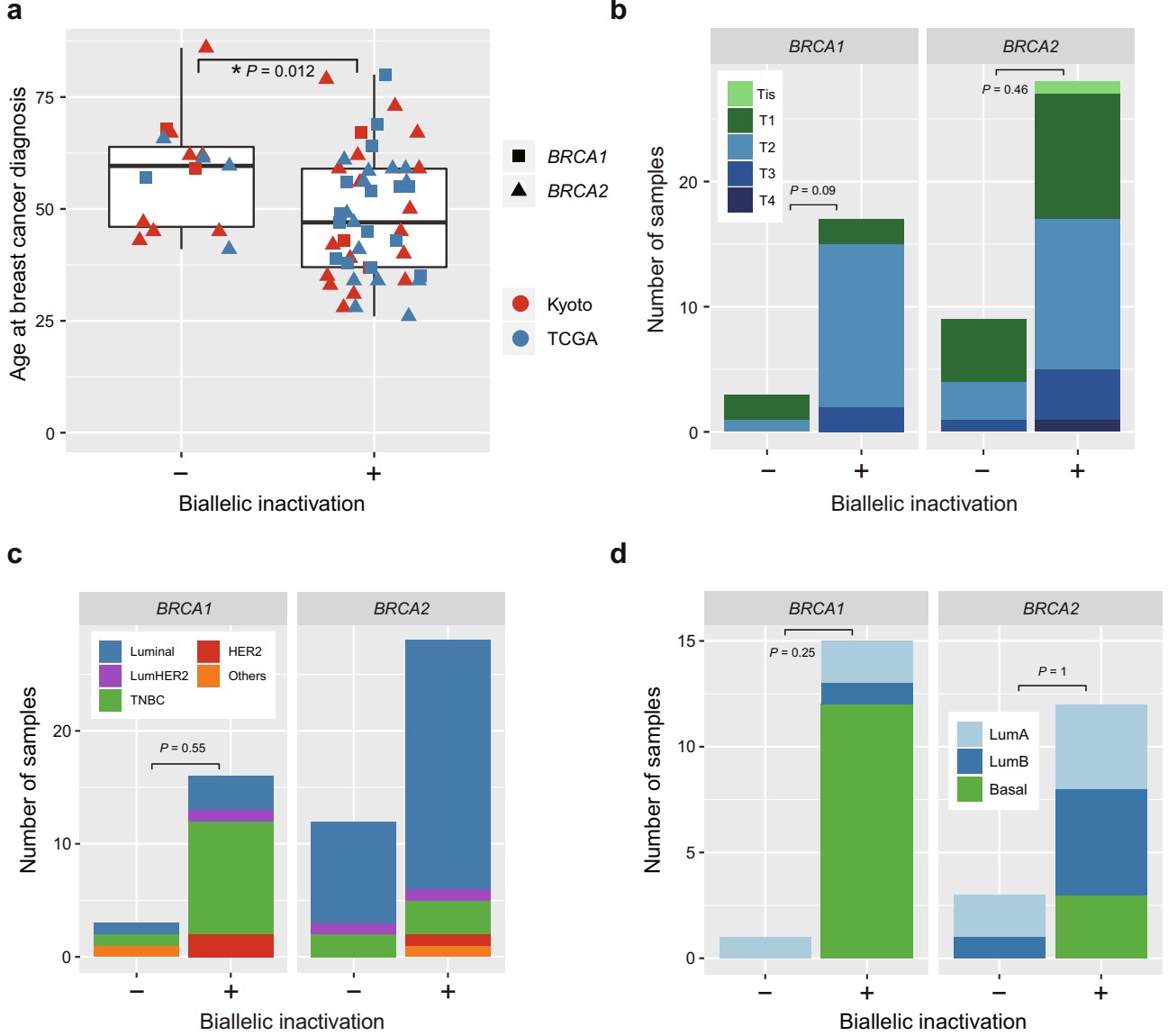

**Fig. 4 Phenotypes of breast tumors harboring germline *BRCA1/2* mutations with and without biallelic inactivation. a** Age at diagnosis of breast cancers harboring germline *BRCA1/2* mutations with ($n = 49$) and without ($n = 15$) biallelic inactivation. The asterisk indicates significance difference (Mann–Whitney $U$ test: $P < 0.05$). **b–d** Distribution of T factor: (**b**), in subtypes based on immunohistochemistry (**c**) in PAM50 mRNA-expression subtypes (**d**) of breast cancers harboring germline *BRCA1/2* variants with and without biallelic inactivation. The difference in frequency of advanced (T2–T4) tumors, triple-negative breast cancer (TNBC), and basal-type tumors between tumors with and without biallelic inactivation were tested by the two-sided Fisher's exact test. Tis carcinoma in situ, LumHER2 luminalHer2, LumA luminalA, LumB luminalB.

In summary, we revealed that breast tumors with pathogenic germline *BRCA1/2* variants show different genetic and clinical characteristics depending on the presence or absence of biallelic inactivation of these genes. Along with the recent data of different impact of mono-allelic and biallelic somatic *TP53* mutations in myelodysplastic syndromes[33], our data highlights the importance of allelic status of cancer driver genes.

## Methods

**Patients and samples**. A total of 2136 breast cancer patients were enrolled in this study, treated between September 2011 and October 2016 at Kyoto Breast Cancer Research Network institutions, consisting of Kyoto University Hospital and 17 affiliated institutions. Among these, 1995 cases fulfilled the following inclusion criteria (Supplementary Fig. 1a): female; sufficient amount of high quality genomic DNA; pathological diagnosis of breast cancer; clinical data of at least one from age of onset, histology, phenotype, grade, clinical stage, past history, and family history.

These cases were collected consecutively with no selection bias. Family history was defined as the presence of one or more first- or second-degree relatives with

breast and/or ovarian cancer. ER, PR, and HER2 status was determined by immunohistochemistry and/or FISH using breast tumor tissue obtained from a core needle biopsy or taken during surgery. For HER2 status, an immunohistochemistry score of 0 and 1+ was considered negative, whereas 3+ was considered positive. Tumors with a score of 2+ were evaluated further by FISH.

Written informed consent was obtained from all participants. The study was reviewed and approved by the Ethics Committees of Kyoto University Graduate School and Faculty of Medicine and Kyoto University Hospital and was performed in accordance with the Declaration of Helsinki (2013 revision).

**Targeted sequencing of pooled DNA samples**. Genomic DNA samples were extracted from peripheral blood samples of patients using a Gentra Puregene Kit (Qiagen). After adjusting the concentration of each genomic DNA sample to 50 ng/μL, each sample from 10 to 20 consecutive patient extractions were combined into one DNA pool[34], generating a total of 106 DNA pools. Pooled DNA samples were analyzed using targeted-capture sequencing of 11 genes implicated in hereditary breast cancer using the SureSelect Custom kit (Agilent Technologies). RNA probes were designed to cover all coding regions and intron–exon boundaries of the 11 breast cancer susceptibility genes. Captured libraries were sequenced on a

HiSeq 2000 (Illumina). To confirm the sensitivity of pooled sequencing, we first generated 2 pools from 40 patients. These 40 patients were individually genotyped and 60 private germline variants in 11 genes were detected. Next, 2 pools were sequenced at 1436–1476×, and 58/60 (97%) of variants were of variant allele frequency (VAF) > 0.005. Subsequently, 106 DNA pools were sequenced. The mean coverage of pooled DNA samples was 1644× (1267×–2316×). Reads were aligned to the reference human genome GRCh37. Germline variations were called using EBCall (Empirical Bayesian mutation Calling, https://github.com/friend1ws/EBCall)[35,36] with for following parameters: (i) a variant allele frequency (VAF) ≥ 0.005; (ii) a P value < 0.01 (by EBCall); (iii) removal of SNPs in ESP, the 1000 genomes project, ExAC and HGVD (http://www.genome.med.kyoto-u.ac.jp/SnpDB/) with a minor allele frequency of ≥0.001; (iv) support from ≥2 reads. Variants were annotated using Annovar (http://annovar.openbioinformatics.org/en/latest/). Three BRCA1/2 mutations already detected in clinical purposes were successfully identified. For each pool with positive variant calls, the variant-positive samples were interrogated by amplicon-sequencing or Sanger sequencing of all samples in the corresponding pool with germline variants. Finally, 101 out of 117 germline mutations were validated. High allele frequencies of variants were supported by both of sequencing chromatograms and the VAFs of amplicon sequencing. Except for the long (20 bp) deletion variants, the VAFs of deep sequencing were ≥0.33, supporting the germline nature of these mutations.

**Variant classification of germline variants**. Truncating mutations (nonsense mutations or frameshift indels) were considered as pathogenic, except for low-risk truncating mutations, such as the K3326X mutations of BRCA2. Missense, synonymous and splice site mutations registered as "pathogenic" or "likely pathogenic" in ClinVar (https://www.ncbi.nlm.nih.gov/clinvar/)[37,38] were also considered as pathogenic variants in this study (Supplementary Fig. 1b).

**Targeted sequencing of tumor samples**. To identify and characterize somatic mutations in tumors from patients with germline variants in BRCA1/2, 36 tumor samples with germline variants in BRCA1/2 and those without pathogenic germline mutations ($n = 35$) were analyzed by target sequencing using a SureSelect system (Agilent). All the tumor samples were collected prior to treatment. For formalin-fixed paraffin-embedded (FFPE) samples, a KAPA Hyper Prep Kit (KAPA Biosystems, Wilmington, MA) was also used before target enrichment. RNA probes were designed to capture 115 genes associated with breast cancer (Supplementary Table 5) and 1275 SNP sites for the measurement of genomic CNs. Based on the allele frequency of mutations and CN changes, we excluded 11 samples with a lower tumor cell fraction from further analysis. Finally, 30 samples with germline variants in BRCA1/2, including frozen tissues ($n = 5$) or FFPE samples ($n = 25$), and 30 tumors without germline mutations were analyzed. The mean coverage of fresh frozen and FFPE samples were 599× (347×–1253×) and 293× (112×–557×), respectively. Somatic mutations were analyzed using EBCall, with the following parameters: (i) removal of SNPs in ESP, the 1000 genomes project, ExAC and HGVD with a minor allele frequency of ≥0.001; (ii) support from ≥5 reads in the tumor; (iii) a VAF ≥ 0.02; (iv) a P value < 0.001 (by EBCall); (v) support from reads mapped to both strands. Variants with a VAF ≥ 0.4 were removed as germline SNPs, except for loss-of-function mutations in tumor suppressor genes and gain-of-function mutations reported in the COSMIC database. Synonymous variants were also excluded as germline variants. Mapping errors were removed by visual inspection on the Integrative Genomics Viewer (http://www.broadinstitute.org/igv/) browser. Finally, mutations in 28 driver genes reported in a previous study[6] (Supplementary Table 5) and hot spot mutations reported in the COSMIC database with ≥10 mutated tumors, including KRAS and CDKN2A mutations, were considered as driver mutations. To confirm the accuracy of mutation calling, we also called single nucleotide variants using MuTect[39] with unmatched control samples with the following parameters: (i) removal of SNPs in ESP, the 1000 genomes project, ExAC and HGVD with minor allele frequency of ≥0.001; (ii) support from ≥5 reads in a tumor; (iii) a VAF ≥ 0.02; (iv) a tumor_alt_fpir_mad > 0. Variants with VAF ≥ 0.4 and synonymous variants were also removed as germline SNPs, except for loss-of-function mutations in tumor suppressor genes and gain-of-function mutations reported in the COSMIC database. We confirmed a high concordance rate between mutations call using both methods, except for a small number of variants with low VAF (Supplementary Fig. 9a).

CN changes were analyzed using CNACS (https://github.com/papaemmelab/toil_cnacs)[40], in which the total number of sequencing reads covering each bait region or SNP probe, and the allele frequency of the heterozygous SNP were used as input data. For the identification of regions with LOH, in addition to deletions and copy-neutral LOHs called by CNACS, we identified gains with LOH based on the estimated tumor purities by total and allele specific (As) CN (Supplementary Fig. 9b). For regions with gain (CN = 3), tumor purity was estimated as follows: (by total CN) Purity = Total CN − 2; (by As CN) Purity = 2 × (1 − As CN)/As CN (gain without LOH); and Purity = 2 × (1 − As CN)/(2 + As CN) (gain with LOH).

We used Control-FREEC[41] with the contaminationAdjustment option, which corrects for contamination by normal cells, to confirm the LOH status of BRCA1/2 loci determined by CNACS. The median of the total and As CN of probes within BRCA1/2, estimated by CNACS and Control-FREEC, were well correlated (Supplementary Fig. 9c). To further confirm the accuracy of CN calling, we called CN changes using SNP array karyotyping for fresh frozen samples ($n = 5$). SNP array-based CN analysis was performed using CNAG software[42,43]. SNP array analysis also provided an almost identical CN profile, including LOH of the BRCA1 and BRCA2 loci (Supplementary Fig. 10), and CNACS detected 57/61 CN alterations identified by SNP array.

**Analysis of TCGA dataset**. Samples subjected to whole-genome amplification were excluded from analysis to accurately define CN changes and SVs. Sequencing data of 829 WES of breast cancer tumors in the TCGA dataset were downloaded. Variants with a VAF ≥ 0.2 in the germline control sample and with <0.001 minor allele frequency in ESP, the 1000 genomes project, ExAC and HGVD were considered as germline mutations. Tumors with pathogenic germline variants of BRCA1/2 ($n = 38$) were defined in the same way as our cohort, and those with somatic BRCA1/2 mutations were not included. Four samples were excluded for the following reasons: low quality of CN data ($n = 3$) and low tumor purity ($n = 1$). Somatic variants were detected using EBCall with following parameters: (i) VAF in tumor samples ≥0.05; (ii) a P value < $10^{-1.3}$ (by Fisher's test); (iii) P value < 0.0001 (by EBCall). Mutational signatures were analyzed using pmsignature[22], and three samples with a high number of artifacts, including TCGA-A2-A0T5, TCGA-A2-A0T6, and TCGA-A7-A0DB were excluded from the analysis. CN changes were also detected from the WES data using CNACS. SVs were analyzed using GenomonSV (https://github.com/Genomon-Project/GenomonSV) as previously reported[44] with additional filters, (i) a frequency in the germline sample <0.02; (ii) a P value < $10^{-1.5}$ (by Fisher's test); (iii) a length of overhang ≥ 100. SVs identified in other samples were also removed as germline variants or errors. Clinical information and PAM50 mRNA subtypes of these samples were extracted from the TCGA database.

**Statistics and reproducibility**. A comparison of categorical variables between mutation carriers and noncarriers was made using the Fisher's exact test or chi-square test where appropriate. For continuous variables, the Mann–Whitney U test was used for group comparisons. P values less than 0.05 were considered statistically significant. The overall survival time for all patients was determined from the date of diagnosis of breast cancer to the time of last follow-up, or death, by examining medical records. Survival was estimated using the Kaplan–Meier product-limit method and differences were tested for statistical significance using the log-rank test. We performed multivariate regression analysis using the Cox proportional hazards model. All analyses were performed using JMP Pro 14.0.0 software.

No statistical methods were used to determine sample size since this is an exploratory study. We enrolled as many patients as possible who provided consent for our study during the enrollment period between September 2011 and October 2016. A total of 2136 breast cancer patients were enrolled in this study.

**Data access**. Targeted sequencing data of 106 pooled DNAs and 60 breast cancer samples have been deposited at the European Genome-phenome Archive (https://www.ebi.ac.uk/ega/) under the accession Nos. EGAS00001004630 and EGAS00001004182, respectively.

**Reporting summary**. Further information on research design is available in the Nature Research Reporting Summary linked to this article.

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

## Acknowledgements

We are grateful to the TCGA Research Network (http://cancergenome.nih.gov/) for providing the data analyzed in this manuscript. We would like to thank all of the Kyoto Breast Cancer Research Network (KBCRN) participants, including Mitsuru Tanaka, Susumu Mashima, Tsuyoshi Tachibana, Shingo Sakata, Fumiaki Yotsumoto, Norimichi Kan, and Nobuhiko Shinkura for their help with providing DNA samples and clinical information of the patients in their hospital. We also thank Maki Nakamura for her technical assistance. This work was supported by Grant-in-Aid for Scientific Research on Innovative Areas from the Ministry of Health, Labor and Welfare of Japan (15H05909) [S.O.]. This research was also partially supported by Astra Zeneca Co., Ltd. We would like to thank Enago (www.enago.jp) for the English language review.

## Author contributions

Y.I.-K., K. Yoshida, N.K.-S., S.O., and M. Toi conceived the project. Y. Shiozawa, H.T., Y. Shiraishi, A.S.-O., Y.N., K.C., and S.M. developed the bioinformatics pipelines of the sequencing data. Y.I.-K. prepared samples and performed the sequencing experiments. Y.I.-K., K. Yoshida, T.N., Y. Shiozawa, Y.T., and Y.I. performed the sequencing data analyses. Y.F. performed SNP array karyotyping. M. Takada and Y. Kataoka helped statistical analyses. M.K., N.S., E.S., M. Takada, M. Torii, H.Y., K.Y., R.O., Y.M., H.K., S.T., A.Y., H.S., and T.I. collected the specimens and clinical information. Y.I.-K., K. Yoshida, S.O., and M. Toi wrote the paper. Y.I.-K. and K. Yoshida generated the figures and tables. S.O. and M. Toi led the entire project. All authors participated in discussions and interpretation of the data and results.

## Competing interests

The authors declare the following competing interests: Masakazu Toi: Research grant: Astra Zeneca, Pfizer, Eisai, Chugai Pharma, Astellas Pharma, Daiichi Sankyo, Taiho Pharmaceutical, C & C Research Laboratories, Ono Pharmaceutical, Shimadzu, Nippon-Kayaku, AFI technology, Japan Breast Cancer Research Group. Consulting or Advisory Role: Daiichi Sankyo, Kyowa Hakko Kirin, Athenex Oncology and Bertis. Honoraria as lecture fee: AstraZeneca, Bayer, C & C Research Laboratories, Chugai Pharma, Daiichi Sankyo, Eisai, Eli Lilly, Genomic Health, Konica Minolta, Kyowa Hakko Kirin, MSD, Novartis, Pfizer, Sanofi, Shimadzu, Taiho Pharmaceutical, Takeda, Yakult. Travel and accommodation expenses: Genomic Health Inst., Eli Lilly and companies. Others: A member of board of directors (no salary) Organization for Oncology and Translational Research, Japan Breast Cancer Research Group and Kyoto Breast Cancer Research Network.
