## [Peer Review File · Communications Biology]

This manuscript has been previously reviewed at another Nature Research journal. This document only contains reviewer comments and rebuttal letters for versions considered at Communications Biology.

Reviewers' comments:

Reviewer #2 (Remarks to the Author):

The revised paper is improved

Reviewer #3 (Remarks to the Author):

The authors present a revised manuscript and have done an outstanding job responding to the prior concerns raised about their manuscript. In particular, it is appreciated that the methods are much more detailed, including the statistical tests, and results from complementary methods are compared. They also have provided information on their new copy number analysis – CNACS. Only relatively minor comments remain to be addressed.

1. As noted in the prior comments, the rate of germline mutations is higher than might be expected in their unselected cohort, particularly given that a third of their patients are over age 66. It is particularly interesting since the CHEK2 founder mutations are of European origin. It would be good to comment on why the rate of mutations in an unselected cohort appears to be much higher than in European-based studies.
2. As mentioned, it is not clear that their survival analysis accounts for co-variables. The revised manuscript's survival curves (supplementary fig. 4) still do not account for these covariates. The univariate analysis they reference in the rebuttal (Supplementary table 6) does not address this point.
3. The authors suggest that the purity of the tumor does not affect the detection of LOH, and that no correction for purity is needed unless there is amplification with LOH. Presumably they argue that because CNACS is using SNPs to call LOH, but it's not stated. Control-FREEC does have an option for tumor purity correction, but it's not clear its used.
4. Although they have denoted which tumors belong to which cohorts (TCGA vs. Kyoto) in fig. 3b and 4a, they appear to have combined the cohorts throughout other panels in figures 3 and 4.
5. Their description of copy number profiles based on BRCA1/2 status is misleading. They revised the sentence, but it is still unclear. Suggested is: "...samples with mono-allelic BRCA1/2 lesions mainly showed copy number gains with rare deletions. In contrast, those with biallelic BRCA1 inactivation showed extensive amplifications with or without LoH, and those with biallelic BRCA2 inactivation were characterized by extensive deletions or copy neutral LoH."
6. The sentence in their discussion, "Tumors with biallelic BRCA1/2 inactivation had unique genetic and clinical features...which were not seen in those with mono-allelic inactivation" is still misleading, as noted in the first review. These unique features were not seen in sporadic tumors, and the tumors with mono-allelic deletion fall somewhere in the middle of that spectrum. There appears to be a dose-dependent effect of BRCA1/2 deletion on tumor characteristics such as CNVs, etc. such that mono-allelic BRCA1/2 deletion positions a tumor to be somewhere between a sporadic tumor and a tumor with biallelic deletion. They should still comment more explicitly that in fact, the only significant difference seen between tumors with mono- and biallelic BRCA1/2 inactivation was age of onset.
7. In their rebuttal, the authors comment that they include TCGA samples with bilallelic somatic hits in BRCA1/2. They do not specify that they include tumors with biallelic somatic mutations anywhere in their manuscript, including the methods. They need to either 1) remove those samples or 2) clearly note which samples are biallelic somatic mutations. The inclusion of these samples biases their results, as they may have different characteristics than tumors associated with germline BRCA1/2 mutations. Additionally, since the somatic tumors are selected for having mutations plus LOH, it biases their overall reports of frequencies. They also mix together TCGA samples and non-TGCA samples in multiple figures, which they should not be doing.
8. The authors need to comment whether tumors were pre-treated with anything prior to collection for the study (TGCA are all untreated primaries, so important if they are combined as well).

Treatment also should be included a covariate in downstream analyses of survival. The inclusion of these data is particularly critical because, as the authors note, many of the germline mutation carriers were unaware of their status. Differences in treatment based on known vs. unknown mutation status could skew the survival results in supplementary fig. 4.

9. "Amplification" is defined here as a copy number > 3 (see methods), but this is an unusually low threshold. Usually it is set at 4 or higher. It would be useful if the authors were able to stratify out copy number gains from amplifications, since their use of the word "amplification" may be overstating the observed copy number changes. In their discussion of supplementary fig. 5, the authors need to be more explicit that differences are in comparison to tumors without a germline mutation. Without this stated explicitly, it sounds like the authors are comparing tumors with mono- vs. biallelic BRCA1/2 inactivation, for which these differences do not achieve significance. In fig. 3b, it could be useful to compare tumors with mono-allelic BRCA1/2 inactivation to tumors without a germline mutation (and note the p-value). This could address the question of whether one BRCA1/2 mutation is "enough" to cause more genome-wide LoH than in a sporadic tumor.

9. In fig. 3b, it could be useful to compare tumors with mono-allelic BRCA1/2 inactivation to tumors without a germline mutation (and note the p-value). This comparison could address the question of whether one BRCA1/2 mutation is "enough" to cause more genome-wide LoH than in a sporadic tumor.

10. In the Discussion (page 7, line 213), the comment that the current set of results conflicts with current findings is not entirely correct. The prior manuscript also found unique BRCA-associated mutational signatures and CNA and SVs (contained within homologous recombination deficiency scores). Essentially the only difference is the lack of an association with age of onset. It's also important to clarify that the current study included biallelic somatic mutations, which the prior manuscript did not.

Minor Comments

1. The manuscript would benefit from some editing of English, as there are several syntax errors, as well as some key statements that are unclear due to sentence structure.
2. "Amplification" is defined here as a copy number > 3 (see methods), but this is an unusually low threshold. Usually it is set at 4 or higher. It would be useful if the authors were able to stratify out copy number gains from amplifications, since their use of the word "amplification" may be overstating the observed copy number changes.
3. In their discussion of supplementary fig. 5, the authors need to be more explicit that differences are in comparison to tumors without a germline mutation. Without this stated explicitly, it sounds like the authors are comparing tumors with mono- vs. biallelic BRCA1/2 inactivation, for which these differences do not achieve significance.
4. Typo in fig. 3d ("with" in the figure legend)
5. Supplementary fig. 5f could use a y-axis label.
6. The sentence "According to the two-hit hypothesis, the frequent presence of tumors with mono-allelic BRCA1 and BRCA2 mutations was rather unexpected..." does not make sense, given that several other groups/papers have observed that LoH is not the rule for BRCA1/2 tumors.
7. The following sentence is confusing. "Of interest, previous studies demonstrated using FISH..." Wouldn't loss of the chromosome with WT BRCA1 and WT TP53 leave the cell with one chromosome 17 allele (bearing both defective genes)?

Rebuttal letter

Reviewers' comments:

Reviewer #2 (Remarks to the Author):

The revised paper is improved

Reply:

We are grateful for the reviewer's comment.

Reviewer #3 (Remarks to the Author):

The authors present a revised manuscript and have done an outstanding job responding to the prior concerns raised about their manuscript. In particular, it is appreciated that the methods are much more detailed, including the statistical tests, and results from complementary methods are compared. They also have provided information on their new copy number analysis; CNACS. Only relatively minor comments remain to be addressed.

Reply:

We thank the reviewer for the positive evaluation.

1. As noted in the prior comments, the rate of germline mutations is higher than might be expected in their unselected cohort, particularly given that a third of their patients are over age 66. It is particularly interesting since the CHEK2 founder mutations are of European origin. It would be good to comment on why the rate of mutations in an unselected cohort appears to be much higher than in European-based studies.

We respectfully disagree with the reviewer. In fact, the rate of germline mutations in our unselected cohort was 5.1%, which is much *lower* than the reported frequency in unselected European cohorts: 8.9-9.2% (The Cancer Genome Atlas Network., *Nature* 2012., Tung et al., *J Clin Oncol.* 2016).

2. As mentioned, it is not clear that their survival analysis accounts for co-variates. The revised manuscript's survival curves (supplementary fig. 4) still do not account for these covariates. The univariate analysis they reference in the rebuttal (Supplementary table 6) does not address this point.

We apologize that in the previous submissions, our survival analysis did not account these co-variates. In response to the reviewer's criticism, we performed multivariate analysis using tumor size, state of lymph node metastasis, age and triple-negative disease as co-variates. In this multivariate analysis, no prognostic impact of pathogenic germline variants or biallelic status of *BRCA1/2* mutations was demonstrated. We presented the results in Supplementary Tables 4 and 8 and described in the main text as follows:

(Page 4; Line 113)

"No prognostic impact of germline mutations was demonstrated for overall or disease-free survivals both in univariate and multivariate analyses."

(Page 6; Line 201)

"The mutation status of *BRCA1/2* or the presence or absence of biallelic involvement of these genes did not affected overall or disease-free survivals both in univariate and multivariate regression analyses."

3. The authors suggest that the purity of the tumor does not affect the detection of LOH, and that no correction for purity is needed unless there is amplification with LOH. Presumably they argue that because CNACS is using SNPs to call LOH, but it's not stated. Control-FREEC does have an option for tumor purity correction, but it's not clear its used.

Clearly, in any platforms, the lower the tumor purity, the lower the sensitivity to detect LOH. CNACS detects LOH by seeing allelic imbalance using SNPs, where the lower the purity, the smaller the allelic imbalance and the lower the sensitivity. On the use of Control-FREEC, we used the contaminationAdjustment option for the correction of contamination of normal cells, whereas CNACS does not need such an explicit correction. CNACS assumes the presence of contamination of normal cells by default. In the revised manuscript, we clearly described these in the method section:

(Page 15; Line 463)

"CNACS, in which the total number of sequencing reads covering each bait region or SNP probe, and the allele frequency of heterozygous SNP were used as input data."

(Page 15; Line 471)

"We used Control-FREEC with the contaminationAdjustment option, which correct for contamination by normal cells."

4. Although they have denoted which tumors belong to which cohorts (TCGA vs. Kyoto) in fig. 3b and 4a, they appear to have combined the cohorts throughout other panels in figures 3 and 4.

We denoted which tumors belong to which cohorts (TCGA vs. Kyoto) in fig. 3b and 4a. However, in other panels in fig. 3 and fig. 4, it makes no sense to discriminate the cohorts each tumor belonged. Thus, we just combined the two cohorts throughout other panels in figures 3 and 4.

5. Their description of copy number profiles based on BRCA1/2 status is misleading. They revised the sentence, but it is still unclear. Suggested is: "...samples with mono-allelic BRCA1/2 lesions mainly showed copy number gains with rare deletions. In contrast, those with biallelic BRCA1 inactivation showed extensive amplifications with or without LoH, and those with biallelic BRCA2 inactivation were characterized by extensive deletions or copy neutral LoH."

We are grateful for the reviewer's suggestion, and agree that our previous description might be misleading. According to the reviewer's suggestion, we corrected that in the revised manuscript as follows:

(Page 5; Line 143)

"samples with mono-allelic *BRCA1/2* lesions mainly showed copy number gains with rare deletions. In contrast, biallelic *BRCA1* inactivation showed extensive gains with or without LOH, and those with biallelic *BRCA2* inactivation were characterized by extensive deletions or copy neutral LOHs."

6. The sentence in their discussion, "Tumors with biallelic BRCA1/2 inactivation had unique genetic and clinical features...which were not seen in those with mono-allelic inactivation" is still misleading, as noted in the first review. These unique features were not seen in sporadic tumors, and the tumors with mono-allelic deletion fall somewhere in the middle of that spectrum. There appears to be a dose-dependent effect of BRCA1/2 deletion on tumor characteristics such as CNVs, etc. such that mono-allelic BRCA1/2 deletion positions a tumor to be somewhere between a sporadic tumor and a tumor with biallelic deletion. They should still comment more explicitly that in fact, the only

significant difference seen between tumors with mono- and biallelic *BRCA1/2* inactivation was age of onset.

We respectfully disagree with the reviewer on the first point; mono-allelic cases did not fall into somewhere between biallelic and no inactivation, but showed a similar phenotype with those with no inactivation (Figure 4c, 4d). In the second point, we agree with the reviewer's comment. In accordance with the reviewer's suggestion, we explicitly described that the only significant difference seen between tumors with mono- and biallelic *BRCA1/2* inactivation was age of onset and attenuate the statement regarding the enrichment of biallelic mutations in triple-negative diseases or basal type histology.

(Page 7; Line 215)

"Although the only significant clinical difference between tumors with mono-allelic and biallelic *BRCA1/2* inactivation was age at onset, the frequency of advanced stage, triple-negative or basal tumors tended to be higher in tumors with biallelic inactivation."

7. In their rebuttal, the authors comment that they include TCGA samples with biallelic somatic hits in *BRCA1/2*. They do not specify that they include tumors with biallelic somatic mutations anywhere in their manuscript, including the methods. They need to either 1) remove those samples or 2) clearly note which samples are biallelic somatic mutations. The inclusion of these samples biases their results, as they may have different characteristics than tumors associated with germline *BRCA1/2* mutations. Additionally, since the somatic tumors are selected for having mutations plus LOH, it biases their overall reports of frequencies. They also mix together TCGA samples and non-TGCA samples in multiple figures, which they should not be doing.

We did not include tumors with somatic *BRCA1/2* mutations in TCGA cohort as tumors with *BRCA1/2* mutations as in our cohort. In the revised manuscript, we clearly described this in the method as follows:

(Page 15; Line 486)

"Tumors with pathogenic germline variants of *BRCA1/2* (n = 38) were defined in the same way as our cohort, and those with somatic *BRCA1/2* mutations were not included."

8. The authors need to comment whether tumors were pre-treated with anything prior to collection for the study (TGCA are all untreated primaries, so important if they are combined as well). Treatment also should be included a covariate in downstream analyses of survival. The inclusion of these data is particularly critical because, as the authors note, many of the germline mutation carriers were unaware of their status. Differences in treatment based on known vs. unknown mutation status could skew the survival results in supplementary fig.4.

All the tumor samples sequenced in this study were collected prior to treatment. We had described this in the method section. For the survival analyses, we were not able to collect the information of treatment for a significant fraction of patients, and, therefore, we were forced to exclude treatment from covariates. However, none of the carriers or doctors were aware of their mutation status, because it was analyzed retrospectively. Therefore, the status of germline mutation was not able to affect treatment decisions.

9. "Amplification" is defined here as a copy number > 3 (see methods), but this is an unusually low threshold. Usually it is set at 4 or higher. It would be useful if the authors were able to stratify out copy number gains from amplifications, since their use of the word "amplification" may be

overstating the observed copy number changes. In their discussion of supplementary fig. 5, the authors need to be more explicit that differences are in comparison to tumors without a germline mutation. Without this stated explicitly, it sounds like the authors are comparing tumors with mono- vs. biallelic BRCA1/2 inactivation, for which these differences do not achieve significance. In fig. 3b, it could be useful to compare tumors with mono-allelic BRCA1/2 inactivation to tumors without a germline mutation (and note the p-value). This could address the question of whether one BRCA1/2 mutation is “enough” to cause more genome-wide LoH than in a sporadic tumor)

Previously, as the reviewer mentioned, we define “amplification” as a copy number ≥ 3 . We agree that “gain” is a more appropriate word to express it, and we corrected “amplification” to “gain” throughout the manuscript. It does not make any changes but just the problem of terminology.

As for Supplementary Figure 5, we additionally described that we have compared tumors with biallelic inactivation of *BRCA1/2* to tumors without germline mutations as follows:

(Page 5; Line 161)

“Of these, the mutational signature caused by deficient HR (Sig_3) was more frequent in tumors with biallelic *BRCA2* inactivation than those without germline mutations,”

(Page 6; Line 165)

“compared to tumors without germline variants, increased occurrence of tandem duplications and deletions (for *BRCA1* inactivation) and deletions (for *BRCA2* inactivation) were observed”

Also, in response to the comment, we added the p-values to show that there were no significant differences between tumors with mono-allelic *BRCA1/2* inactivation and those without germline mutations in Figure 3b.

10. In the Discussion (page 7, line 213), the comment that the current set of results conflicts with current findings is not entirely correct. The prior manuscript also found unique BRCA-associated mutational signatures and CNA and SVs (contained within homologous recombination deficiency scores).

We agree with the reviewer and revised the text as follows:

(Page 7; Line 218)

“The correspondence of biallelic *BRCA1/2* inactivation with earlier age of onset conflicts with results of a previous study²².”

Essentially the only difference is the lack of an association with age of onset.

We agree with the reviewer. According to the reviewer’s suggestion, we explicitly described that the only significant difference seen between tumors with mono- and biallelic BRCA1/2 inactivation was age of onset and attenuate the statement regarding the enrichment of biallelic mutations in triple-negative diseases or basal type histology.

(Page 7; Line 215)

“Although the only significant clinical difference between tumors with mono-allelic and biallelic *BRCA1/2* inactivation was age at onset, the frequency of advanced stage, triple-negative or basal tumors tended to be higher in tumors with biallelic inactivation.”

It's also important to clarify that the current study included biallelic somatic mutations, which the prior manuscript did not.

Also, as described in the answer to the comment#7, we did not include tumors with somatic *BRCA1/2* mutations as tumors with *BRCA1/2* mutations.

Minor Comments

1. The manuscript would benefit from some editing of English, as there are several syntax errors, as well as some key statements that are unclear due to sentence structure.

According to the reviewer's suggestion, the revised manuscript has been fully edited by the proofreader (Enago).

2. "Amplification" is defined here as a copy number > 3 (see methods), but this is an unusually low threshold. Usually it is set at 4 or higher. It would be useful if the authors were able to stratify out copy number gains from amplifications, since their use of the word "amplification" may be overstating the observed copy number changes.

This comment is the same as a major comment #9.

3. In their discussion of supplementary fig. 5, the authors need to be more explicit that differences are in comparison to tumors without a germline mutation. Without this stated explicitly, it sounds like the authors are comparing tumors with mono- vs. biallelic *BRCA1/2* inactivation, for which these differences do not achieve significance.

This comment is also the same as a major comment #9.

4. Typo in fig. 3d ("with" in the figure legend)

We apologize for this. We have corrected this typo.

5. Supplementary fig. 5f could use a y-axis label.

In response to the reviewer's request, we added "Frequency" as a y-axis label of Supplementary Figure 5f.

6. The sentence "According to the two-hit hypothesis, the frequent presence of tumors with mono-allelic *BRCA1* and *BRCA2* mutations was rather unexpected..." does not make sense, given that several other groups/papers have observed that LOH is not the rule for *BRCA1/2* tumors.

We agree with the reviewer's comments. We corrected as follows:

(Page 7; Line 212)

"In line with the previous reports^{6,23,24}, tumors with mono-allelic *BRCA1* and *BRCA2* mutations were also frequently observed in our cohort,"

7. The following sentence is confusing. "Of interest, previous studies demonstrated using FISH... Wouldn't loss of the chromosome with WT *BRCA1* and WT *TP53* leave the cell with one chromosome 17 allele (bearing both defective genes)?"

As the reviewer thought, we intended to mention about tumors which lost wild-type *BRCA1* and *TP53* through LOH of chromosome 17.

In the revised manuscript, we clearly described this as follows:

(Page 8; Line 247)

“tumors with biallelic inactivation of both *BRCA1* and *TP53* have demonstrated that *TP53* mutation occurs before LOH of the intact chromosome 17 as there remained cells with two chromosome 17 alleles and mutated *TP53*”

REVIEWERS' COMMENTS:

Reviewer #3 (Remarks to the Author):

The authors have thoroughly responded to the multiple comments and improved the readability of the manuscript substantially.